# Bactericidal Activity of Non-Cytotoxic Cationic Nanoparticles against Clinically and Environmentally Relevant *Pseudomonas* spp. Isolates

**DOI:** 10.3390/pharmaceutics13091411

**Published:** 2021-09-06

**Authors:** Anna Maria Schito, Gabriella Piatti, Debora Caviglia, Guendalina Zuccari, Alessia Zorzoli, Danilo Marimpietri, Silvana Alfei

**Affiliations:** 1Department of Surgical Sciences and Integrated Diagnostics (DISC), University of Genoa, Viale Benedetto XV, 6, I-16132 Genova, Italy; amschito@unige.it (A.M.S.); gabriella.piatti@unige.it (G.P.); Cavigliad86@gmail.com (D.C.); 2Department of Pharmacy, University of Genoa, Viale Cembrano, 16148 Genoa, Italy; zuccari@difar.unige.it; 3Stem Cell Laboratory and Cell Therapy Center, IRCCS Istituto Giannina Gaslini, Via Gerolamo Gaslini 5, 16147 Genova, Italy; alessiazorzoli@gaslini.org (A.Z.); danilomarimpietri@gaslini.org (D.M.)

**Keywords:** multi-drug-resistant bacteria, Gram-negative opportunistic pathogens, nosocomial infections, plant infections, bacterial food spoilage, *Pseudomonas* human pathogens, *P. syringae* plant pathogen, environmental *Pseudomonas* isolates, lysine-modified cationic dendrimer, nanoparticles

## Abstract

Difficult-to-treat bacterial infections caused by resistant human and plant pathogens severely afflict hospitals, and concern the agri-food sectors. Bacteria from the Pseudomonadaceae family, such as *P. aeruginosa, P. putida, P. fluorescens,* and *P. straminea,* can be responsible for severe nosocomial infections in humans. *P. fragi* is the major cause of dairy and meat spoilage, while *P. syringae* can infect a wide range of economically important plant species, including tobacco, kiwi, and tomato. Therefore, a cationic water-soluble lysine dendrimer (G5-PDK) was tested on several species of *Pseudomonas* genus. Interestingly, G5-PDK demonstrated variable minimum inhibitory concentrations (MICs), depending on their pigment production, on *Pseudomonas aeruginosa* (1.6-> 6.4 µM), MICs = 3.2–6.4 µM on *P. putida* clinical isolates producing pyoverdine, and very low MICs (0.2–1.6 µM) on strains that produced non-pigmented colonies. Time-kill experiments established the rapid bactericidal activity of G5-PDK. In the cytotoxicity experiments on human keratinocytes, after 4 h of treatment with G5-PDK at concentrations 16–500 × MIC, more than 80% of viable cells were observed, and after 24 h, the selectivity indices were maintained above the maximum value reported as acceptable. Due to its proven bactericidal potency and low cytotoxicity, G5-PDK should be seriously considered to counteract clinically and environmentally relevant *Pseudomonas* isolates.

## 1. Introduction

Several genera of non-glucose-fermenting Gram-negative bacteria belonging to the class of Gammaproteobacteria are medically, ecologically, and scientifically important. These microorganisms can live in different terrestrial and marine environments and in various extreme conditions, such as hydrothermal vents. They are generally widely distributed in plant materials, soil, and water, where they play several important roles, such as in the production of antibiotics or the reduction of environmental pollution, but can also adapt to living in healthcare environments. Some species of genus *Pseudomonas* are dangerous opportunistic pathogens for humans. They can colonize the damaged mucosal surface of humans, thus producing infections that are difficult to treat, also due to their remarkable ability to develop resistance to various drugs [1,2,3]. Furthermore, bacteria belonging to the *P. syringae* species are phytopathogens that can infect over 50 different plant species, with a devastating economic impact [4]. The most common *Pseudomonas* species isolated from human clinical specimens fall into the fluorescent *Pseudomonas* group, which includes *P. aeruginosa*, *P. putida*, *P. fluorescens*, and *P. straminea*. Other species, such as *P. stutzeri*, *P. mendocina*, *P. alcaligenes*, *P. pseudoalcaligenes*, *P. luteola*, and *P. oryzihabitans*, are isolated less frequently [5]. The environmental isolates of *P. syringeae* species can infect economically important vegetables such as kiwi, tomatoes, beans, rice, and tobacco; and several tree species such as European horse chestnut, olive, and cherry trees. These pathogenic strains vary in their virulence characteristics and have distinct genes that code for different toxins, and can develop resistance to antibiotics such as streptomycin, which has been adopted for years to limit contaminations of plants by *P. syringae* [6]. Non-pathogenic plant-associated species can act as plant growth promoters. *P. fluorescens* produce insecticides and therefore can be used as biocontrol agents capable of promoting plant health through the antagonism of phytopathogens. Furthermore, the bacteria of this species are characterized by exceptional nutritional versatility that allows them to use a wide range of compounds as carbon sources, including dangerous environmental contaminants. *P. putida* is another typical example of bacterial species endowed with a wide range of biodegradative capabilities, which allow it to use unusual carbon sources, such as toxic organic waste (e.g., petroleum and aromatic hydrocarbons, toluene), thus playing an important role in bioremediation [7].

However, all species of the fluorescent Pseudomonas group can become important and dangerous human opportunistic pathogens, and can represent a critical cause of morbidity and mortality in immunocompromised patients [8]. *P. aeruginosa* is the most alarming opportunistic human pathogen, frequently isolated from hospitalized patients [9], and it is considered the major cause of burn and eye infections, and of severe lung diseases in cystic fibrosis patients [10]. Since the clinical relevance of *P. aeruginosa* already has been extensively reported and is well known, herein we described in detail other less-known species of the *Pseudomonas* genus considered in this study. *P. fluorescens*, unlike *P. aeruginosa*, is generally considered to be of low virulence and an infrequent cause of human infection [11], thus often being considered rather as a saprophytic rhizobacterium, with no pathogenic potential. Nevertheless, the frequent isolation of strains belonging to this species from clinical samples indicates the emergence of new strains that are particularly virulent and resistant to the immune system and drug treatment. *P. fluorescens* has been isolated in hospitals from water or humid environments, such as sinks, drains, toilet linen, and objects, as well as from patients, where they were part of the normal epidermal and oropharyngeal flora [12,13,14]. Most of these isolates are psychotropic, being able to grow at refrigeration temperatures up to 4 °C used to store blood products, distilled water, and disinfectants. *P. fluorescens* bacteria can alter patient’s food, biological reagents, and blood-injecting solutes or derivatives, regardless of whether they are stored under refrigeration [15,16,17,18,19,20,21,22,23]. Capable of producing many enzymes, such as proteases, lipases, and lecithinases, *P. fluorescens* can alter the organoleptic characteristics of foods, representing an important cause of food spoilage. Since the growth of *P. fluorescens* occurs at low temperatures, identification of bacteria of this species is difficult at the standard microbiology laboratory incubation temperature of 37 °C, thus making their early diagnosis more difficult than that of *P. aeruginosa* [9]. *P. fluorescens* has been involved in cases of urinary tract infections in immunocompromised patients and of the respiratory tract in a patient with lung cancer [24,25]. It is also a cause of septicaemia, related to blood transfusion [26,27] or catheters in cancer patients [11]. Outbreaks of *P. fluorescens* bacteremia have been reported in cancer outpatients, using syringes with intravenous heparin catheter flushing over implantable venous ports [28]. Outbreaks of pseudo-bacteremia from contamination of disinfectants and blood collection tubes have been described [29,30]. Isolates of *P. fluorescens* have been involved in cases of peritonitis in patients on peritoneal dialysis, lethal liver infections, etc. [31,32,33,34,35]. *P. fluorescens* can also resist many antibiotics and antiseptics, and when grown at low temperatures, it is also able to induce expression of proinflammatory cytokines, beta-defensin 2, and the intercellular adhesion molecule-1; while when grown at 37 °C, it produces biofilm.

There is evidence that nosocomial plagues by *P. fluorescens* bacteria often have water sources [36]. In this regard, a sudden increase in rates of *P. fluorescens* isolation from pharyngeal swabs from hematology patients in the bone marrow transplant unit was observed by Vanessa Wong and co-workers in their teaching hospital [37]. These isolates were the cause of febrile neutropenia, resistant to treatment with chloramphenicol, ampicillin, narrow-spectrum cephalosporins, and other cephalosporins, including cefuroxime, cefotaxime, cefmenoxime, and cefsulodin.

*P. putida* is a saprotrophic soil bacterium that, based on 16S rRNA analysis, was taxonomically confirmed to be a “sensu stricto” *Pseudomonas* species and was placed, along with several other species, in the *P. putida* group, including *P. alkylphenolia*, *P. monteilii*, *P. cremoricolorata*, *P. fulva*, *P. parafulva*, *P. entomophila*, *P. mosselii*, and *P. plecoglossicida* [38,39].

Wild-type strains of *P. putida* can be used for soil bioremediation, to remedy naphthalene contamination, to convert styrene oil into PHA biodegradable plastic, and for the effective recycling of otherwise not biodegradable expanded polystyrene [40,41,42].

*P. putida* has shown potential biocontrol properties, as it behaves as an effective antagonist of Pythium and Fusarium [43,44]. Strains of *P. putida* can be exploited for the degradation of caffeine into carbon dioxide and ammonia [45,46], while their genetic manipulation has allowed them to be used in the synthesis of numerous pharmaceutical and agricultural organic compounds [47].

Mainly because *P. putida* lacks key genes that *P. aeruginosa* uses to cause disease in humans and cannot degrade cell membranes or release toxins, it is considered safe. Despite this, many studies have reported that *P. putida* is the cause of severe illness in people with compromised immune systems, including a lethal case of bacteremia [48]. A recent study developed to analyze the risk factors, clinical characteristics, and antimicrobial resistance of *P. putida* isolated from Tongji Hospital in Wuhan (China) found high rates (75%) of drug resistance in *P. putida* isolates, particularly to trimethoprim/sulfamethoxazole (97.7%), aztreonam (88.6%), minocilin (74.3%), and ticarcillin/clavulanic acid (72.7%) [49].

Like *P. fluorescens*, *P. putida* can cause, especially in immune-compromised adult patients, various types of serious infections, such as bacteremia, pneumonia or peritonitis, meningitis, septic arthritis, endocarditis, and osteomyelitis [50,51,52,53,54,55], or more localized infections, such as sinus or urinary tract infections.

Although infrequently, *P. putida*-borne infections have also been observed among children [51].

*P. straminea* is a typical endophytic bacterium, not pathogenic in humans nor in plants, with an interesting antagonist activity against fungal pathogens [50]. *P. lundensis* and *P. fragi*, which also live in the aqueous environment, are the main species encountered on carcasses after slaughter and in dairy products, causing deterioration, increasing food waste, and negatively affecting the food sector [56].

Globally, *P. fluorescens*, *P. putida*, and *P. syringae* are potential pathogens for both humans and various types of plants, such as tomatoes, kiwis, and tobacco, and are also capable of developing drug resistance. For these species and for *P. fragi*, which are capable of causing food putrefaction with consequent food waste, there is an urgent need to develop new antibacterial agents that are unlikely to induce resistance.

The current situation and update concerning the available antibiotics to counteract isolates of *Pseudomonas* genus is alarming. The Antimicrobial Availability Task Force of the Infectious Diseases Society of America, the FDA, and other organizations have highlighted the urgent need to develop new antibiotics with activity against Gram-negative organisms, including *P. aeruginosa* [57]. Unfortunately, novel agents with activity against *P. aeruginosa* may not be available in the next 9 to 11 years [57]. In the meantime, colistin (polymyxin E) is often the only available active antibiotic, and it is increasingly used as the last line of therapy against Gram-negative “superbugs”. Indeed, a study on a total of 168 patients having either community-acquired (54.2%) or nosocomial-acquired (45.8) pneumonia caused by non-resistant and MDR *P. aeruginosa* evidenced a very high resistance to fosfomycin. A similar resistance pattern was seen with ciprofloxacin, levofloxacin, ceftazidime, piperacillin, imipenem, piperacillin and tazobactam, tobramycin, gentamicin, and meropenem. An elevated resistance of both *P. aeruginosa* and MDR *P. aeruginosa* was found for cefepime and amikacin [58]. Neither pathogen was resistant to colistin [58], but cases of *P. aeruginosa* isolates resistant to colistin by arnB deletion and/or through the addition of l-Ara4N to lipid A have been documented [59]. In this alarming scenario, consisting of antibiotics no longer active and therapeutic options that are not entirely safe and the activity of which probably will not endure, it is necessary to search for new antibacterial agents, active particularly against *Pseudomonas* species and especially *P. aeruginosa.*.

To this end, in this study, the antibacterial and bactericidal activity of a biodegradable synthetic cationic dendrimer, peripherally functionalized with lysine (G5-PDK), was assessed on several clinical or environmental isolates of *P. aeruginosa, P. fluorescence, P. putida, P. straminea, P. syringae,* and *P. fragi* species, with interesting and promising results. A strain of *P. oleovorans* was also considered in this study that, although it is a methylotrophic bacterium commonly isolated from water–oil emulsions used as lubricants and cooling agents for metal cutting, was unexpectedly isolated from a hospitalized individual. Furthermore, to evaluate the possible clinical application of G5-PDK, its cytotoxicity on human keratinocyte cells was evaluated.

## 2. Materials and Methods

### 2.1. Chemicals and Instruments

The fifth-generation biodegradable cationic dendrimer G5-PDK was prepared according to a recently reported synthetic procedure [60]. Experimental details and characterization data are available in the Appendix A (Appendix A, including Appendix A; and Appendix A). A stylized representation of the structure of G5-PDK is presented in Appendix A. All reagents and solvents were purchased from Merck (formerly Sigma-Aldrich, Darmstadt, Germany) and were purified by standard procedures. Melting points and boiling points are uncorrected. Chemical materials and methods, as well as instruments used for the physicochemical characterization of G5-PDK, can be found in our previous work [60].

### 2.2. Microbiology

#### 2.2.1. Microorganisms

A total of 49 strains belonging to the non-fermenting Gram-negative family (1 *S. maltophylia* and 48 *Pseudomonas)* were used in this study. Except for an ATCC 27853 *P. aeruginosa* strain, all were clinical or environmental isolates belonging to a collection obtained from the School of Medicine and Pharmacy of the University of Genova, identified by VITEK^®^ 2 (Biomerieux, Firenze, Italy) or matrix-assisted laser desorption/ionization time-of-flight (MALDI-TOF) mass spectrometric technique (Biomerieux, Firenze, Italy). Of the tested *Pseudomonas*, 32 were *P. aeruginosa*, (1 was an ATCC strain, while 31 were clinical isolates, including 4 non-pigmented strains, 25 from yellow to blue-green-pigmented and 2 brown-pigmented). Five strains were *P. fluorescens* environmental isolates, 5 were *P. putida* (4 clinical isolates and 1 environmental isolate), 3 were *P. straminea*, 1 was a *P. fragi*, and 1 was a *P. syringae* environmental isolates, while 1 was a *P. oleovorans* clinical isolate.

#### 2.2.2. Determination of the MIC

To investigate the antimicrobial activity of G5-PDK on the 49 isolates, their minimal inhibitory concentrations (MICs) were determined by following the microdilution procedures detailed by the European Committee on Antimicrobial Susceptibility Testing (EUCAST) [61].

Briefly, overnight cultures of bacteria were diluted to yield a standardized inoculum of 1.5 × 10^8^ CFU/mL. Aliquots of each suspension were added to 96-well microplates containing the same volumes of serial 2-fold dilutions (ranging from 1 to 128 μg/mL) of G5-PDK to yield a final concentration of about 5 × 10^5^ cells/mL. The plates were then incubated at 37 °C. After 24 h of incubation at 37 °C, the lowest concentration of G5-PDK that prevented a visible growth was recorded as the MIC. All MICs were obtained in triplicate, the degree of concordance in all the experiments was 3/3, and the standard deviation (±SD) was zero.

#### 2.2.3. Killing Curves

Killing curve assays for G5-PDK were performed on representative isolates of *P. aeruginosa, P. putida,* and *P. fluorescens* as previously reported [62]. Experiments were performed over 24 h at G5-PDK concentrations of four times the MIC for all strains.

A mid logarithmic phase culture was diluted in Mueller–Hinton (MH) broth (Merck, Darmstadt, Germany) (10 mL) containing 4 × MIC of the dendrimer to give a final inoculum of 1.0 × 10^5^ CFU/mL. The same inoculum was added to cation-supplemented Mueller–Hinton broth (CSMHB) (Merck, Darmstadt, Germany), as a growth control. Tubes were incubated at 37 °C with constant shaking for 24 h. Samples of 0.20 mL from each tube were removed at 0, 2, 4, 6, and 24 h, diluted appropriately with a 0.9% sodium chloride solution to avoid carryover of G5-PDK being tested, plated onto MH plates, and incubated for 24 h at 37 °C. Growth controls were run in parallel. The percentage of surviving bacterial cells was determined for each sampling time by comparing colony counts with those of standard dilutions of the growth control. Results have been expressed as log10 of viable cell numbers (CFU/mL) of surviving bacterial cells over a 24 h period. Bactericidal effect was defined as a 3 log10 decrease of CFU/mL (99.9% killing) of the initial inoculum. All time-kill curve experiments were performed in triplicate.

### 2.3. Evaluation of G5-PDK Cytotoxicity 

#### 2.3.1. Cell Culture

Human skin keratinocytes cells (HaCaT), obtained thanks to a generous gift from the Laboratory of Experimental Therapies in Oncology, IRCCS Istituto Giannina Gaslini (Genoa, Italy), were grown as a monolayer in RPMI 1640 medium supplemented with 10% fetal bovine serum (v/v), 1% penicillin-streptomycin, 1% glutamine (Euroclone S.p.A., Milan, Italy), cultured in T-25 cm^2^ plastic flasks (Corning, Corning, NY, USA) and maintained at 37 °C in a 5% CO_2_ humidified atmosphere. Cells were tested and characterized at time of experimentation as previously described [63].

#### 2.3.2. Viability Assay

HaCaT cells were seeded in 96-well plates (at 4 × 103 cells/well) in complete medium and cultured for 24 h. The seeding medium was removed and replaced with fresh complete medium that had been supplemented with increasing concentrations of dendrimer (0 μM, 1 μM, 5 μM, 10 μM, 25 μM, 50 μM, 75 μM, 100 μM). Cells (quadruplicate samples for each condition) were then incubated for an additional 4, 12, or 24 h. The effect on cell growth was evaluated by a fluorescence-based proliferation and cytotoxicity assay (CyQUANT^®^ Direct Cell Proliferation Assay, Thermo Fisher Scientific, Life Technologies, MB, Italy) according to the manufacturer’s instructions. Briefly, at the selected times, an equal volume of detection reagent was added to the cells in culture and incubated for 60 min at 37 °C. The fluorescence of the samples was measured using the monochromator-based M200 plate reader (Tecan, Männedorf, Switzerland) set at 480/535 nm. The experiments were carried out at least three times and each sample was run in quadruplicate. Once obtained the results, we simultaneously considered both concentrations and cell viability data for each exposure time to detect the presence of outliers. For each time, we obtained a 2 × 7 matrix (14 variables) on which we performed the principal component analysis (PCA) using the PAST software (Paleontological Statistics software package for education and data analysis), downloadable free of charge online at: https://past.en.lo4d.com/windows (accessed on 31 August 2021). Once we removed the outliers, residual data from cytotoxicity experiments at 12 and 24 h of exposure were used to obtain the LD_50_ values. The LD_50_ value for 24 h of exposure was in turn used to determine the selectivity indices (SIs). In microbiology, the SI value is given by Equation (1), and provides a measure of the selectivity of the antimicrobial agent for bacteria rather than for mammalian cells:(1)$$SI=LD_{50}/MIC$$
where LD_50_ is the dose needed to kill 50% of the cells.

### 2.4. Statistical Analyses

Data are expressed as means ± SD. Statistical significance of differences were determined by a Student’s *t*-test (for differences vs. control), and by one-sample *t*-test using PAST for differences among determinations. Differences were considered statistically significant when *p* < 0.05. As previously reported, concerning MIC and MBC values, experiments were performed in triplicate, the concordance degree was 3/3, and ± SD was zero.

## 3. Results

### 3.1. Synthesis and Spectrophotometric Characterization of Lysine-Modified Cationic Dendrimer G5-PDK

#### 3.1.1. Synthesis of the Uncharged Fifth-Generation Polyester-Based Inner Scaffold of G5-PDK (G5-PD-OH)

Performing previously reported procedures [64,65,66,67,68,69], starting from the AB_2_ monomer known as *bis*-hydroxymethyl propanoic acid *bis*-HMPA, we firstly prepared the fifth-generation dendron D5-A-COOH, the structure of which is shown in Appendix A, available in the Appendix A. Subsequently, according to Appendix A, we synthetized the uncharged dendrimer G5-PD-OH, having 64 peripherals OH groups [67].

The synthetic intermediates and the final compound G5-PD-OH were spectrophotometrically characterized by FTIR and NMR analyses, as presented in the Appendix A. The acquired spectra agreed with reported data, thus confirming all the structures [67]. The molecular weight (MW) of G5-PD-OH was calculated on the base of its structure and was furtherly validated by elemental analysis. The results are available in Appendix A.

#### 3.1.2. Synthesis of Lysine-Modified Boc-Protected Dendrimer G5-PD-BK

G5-PD-BK was obtained according to a recently reported procedure [60] available in Appendix A and presented in Appendix A. G5-PD-BK was spectrophotometrically characterized by FTIR and NMR analyses that confirmed its structure. The MW of G5-PD-BK was calculated based on its structure, and was further confirmed by elemental analysis (Appendix A.

#### 3.1.3. Acidic Deprotection of G5-PD-BK to Obtain G5-PDK * 128 HCl

G5-PDK was obtained as recently reported [60], and experimental details have been included in Appendix A. G5-PDK was spectrophotometrically characterized by FTIR and NMR analyses that confirmed its structure. The MW of G5-PDK was calculated based on its structure, and was further validated by elemental analysis (Appendix A) and by volumetric titration. Potentiometric titrations were performed to assess the protonation profile of G5-PDK (Appendix A, and Appendix A), while its particle size (Z-ave, nm), polydispersity index (PDI), and Z-potential (ζ-p, mV) were determined by dynamic light scattering (DLS) analysis (Appendix A). A detailed discussion of the characterization results is available in Alfei et al. (2021) [60]. To facilitate the readers, the main physicochemical properties of G5-PD-OH and G5-PD-BK have been included in the following Table 1, while those of G5-PDK are in Table 2.

### 3.2. Antibacterial Properties

#### 3.2.1. MICs of G5-PDK

Preliminary investigations of relevant multi-drug-resistant (MDR) representatives of both Gram-positive and Gram-negative bacteria, performed to screen for the antibacterial effects of G5-PDK, had shown that it was inactive towards *Enterococci*, *Staphylococci*, and *Enterobacteriaceae* [60]. On the contrary, G5-PDK proved to possess a remarkable antibacterial activity against *Acinetobacter baumannii* (Appendix A). Considering the clinical relevance of *A. baumannii*, and the limited number of studies on the antibacterial activity of cationic dendrimers on this species, we decided to extend our investigations to different clinical isolates of *A. baumannii*, and of other species of the genus *Acinetobacter.* Regarding this, we recently published the self-biodegradability and the rapid bactericidal activity of G5PDK against *A. baumannii, A. johnsonii, A. junii, A. pittii,* and *A. ursingii* (Appendix A) [60]. As G5-PDK is active against *A. baumannii,* we its antibacterial effects against other relevant non-fermenting Gram-negative genera such as *Stenotrophomonas* and *Pseudomonas*. Interestingly, while G5-PDK was practically inactive against the initially tested isolates of *P. aeruginosa* and *S. maltophylia*, it showed very low MICs against other *Pseudomonas* species belonging to the fluorescens group (Table 3).

To understand the reason for such great differences in antibacterial activity against bacterial species of the same genus and the same phylogenetic group, we have deepened the study of the antibacterial activity of G5-PDK against different species of the genus *Pseudomonas*. To this end, we included in the study many *P. aeruginosa* isolates, and other clinically and economically relevant species of the *Pseudomonas* genus rarely considered that, like *P. aeruginosa*, require new antibacterial treatments. MIC values for G5-PDK were obtained analyzing a total of 48 strains belonging to the *Pseudomonas* genus, including 32 strains of *P. aeruginosa*, 5 of *P. fluorescens*, 5 of *P. putida*, 3 of *P. straminea,* and 1 representative each of *P. oleovorans*, *P. fragi,* and *P. syringae* species. The MICs observed for G5-PDK against the isolates tested in this study are reported in Table 4, which also shows the selectivity indices of G5-PDK for each strain, determined using the LD_50_ of G5PDK obtained from the cytotoxicity experiment on human keratinocytes (Ha-CaT).

Curiously, although all strains tested had clinical origin, very different results were obtained concerning the antibacterial activity of G5-PDK against the isolates of *P. aeruginosa* species. Particularly, G5-PDK displayed very high MICs (>6.4 µM) on the 78.1% of *P. aeruginosa* strains reported in Table 4, which, as shown in Figure 1, produced yellowish, yellowish-green, green-blue or blue-pigmented colonies (strains with an asterisk (*) in Table 4). The only exception to this trend was represented by strain 256, which, although strongly pigmented, displayed MIC = 6.4 µM, equal to those of non-pigmented isolates.

Indeed, the non-pigmented *P. aeruginosa* isolates (strains 2, 4, 9, 12; Table 4, Figure 1), showed lower MICs (6.4 µM) then the colored ones. However, two exceptions to this behavior were observed, represented by strains 14 and 17, which, although not pigmented, showed MIC > 6.4 µM. Finally, much lower MICs (1.6 µM) were observed for the brown-pigmented *P. aeruginosa* strains 244 (Figure 1) and 432 (Table 4).

Except for only three out of 32 strains (9%), these results can be explained by considering the different pigments that *P. aeruginosa* can produce, which can significantly influence the virulence of the genus *Pseudomonas*, but also their tolerance to cationic antibiotics, as noted in this study. In fact, *P. aeruginosa* can produce four types of pigments, including pyocyanin, pyoverdine, pyorubrin, and pyomelanin. While pyorubrin is a red pigment uncommonly produced by *P. aeruginosa* [70,71] that gives colonies a red color, pyocyanin (blue-green pigment) and pyoverdine (yellow-green pigment) are produced by the most strains of *P. aeruginosa* [72]. These pigments give the colonies yellowish, yellow-green, green-blue, or blue colors, such as those we observed for most of the *P. aeruginosa* isolates in this study (78%), which showed high tolerance to G5-PDK (MICs > 6.4 µM; Table 4, Figure 2).

Particularly, pyoverdine is both a virulence factor with many capacities to damage the infected cells and a siderophore involved in the acquisition of iron, an essential element for the functions and life of *Pseudomonas* genus [73]. One of the three domains that make up the structure of pyoverdine is a peptide (Figure 2), the composition of which is strain-dependent, but includes polar amino acids rich in protonable guanidine and amine groups [74,75]. Overall, the structure of pyoverdine resembles that of natural antibacterial peptides (AMPs), which are cationic molecules. Therefore, its secretion in the culture medium creates a cationic barrier capable of rejecting other cationic molecules, such as G5-PDK, preventing its interaction with the bacterial surface and canceling its antibacterial effects, thus justifying the low activity of G5-PDK towards the yellow to yellow-green isolates of *P. aeruginosa* observed in this study. Furthermore, according to the biosynthesis from *P. aeruginosa* [76], the phenazine structure of pyocyanin, at blood pH (pH = 7.4), also is cationic (Figure 2). This molecule, in addition to being an active redox secondary metabolite acting as a virulence factor [76,77], can reject other cationic molecules and hinder the interaction of G5-PDK with the surface of pathogens, just as pyoverdine can, thus further helping in this study to reduce the susceptibility of pyocyanin-producing strains of *P. aeruginosa* to G5-PDK. In fact, very recently has been shown that pyocyanin activates mechanisms that give pyocyanin-producing strains of *P. aeruginosa* high tolerance to several synthetic clinical antibiotics, thus favoring the evolution of antibiotic resistance [78]. In contrast, pyomelanin is a reddish-brown pigment produced by only a few rare strains of *P. aeruginosa* [79] that appear brown in color, such as the *P. aeruginosa* isolates in this study, which showed high susceptibility to P5-PDK (Table 4 and Figure 3).

Unlike pyoverdine and pyocyanin, pyomelanin is a negatively charged pigment that, when secreted, creates an anionic environment. This event, by strengthening the intrinsic negative charge of the outer membrane of the strains that produce pyomelanin, promotes the adsorption of cationic antibacterial agents, such as G5-PDK. This may explain the high susceptibility of the pyomelanin-producers *P. aeruginosa* in this study (strains 244 and 432) to G5PDK.

Regarding non-pigmented *P. aeruginosa* strains (Figure 1), due to the absence of any type of pigment, the interaction of G5-PDK with the bacterial surface was neither hindered nor promoted, and MICs = 6.4 µM were observed (Table 4). Concerning the other *Pseudomonas* species, the most susceptible strains were the phytopathogen *P. syringae* (MICs = 0.2 µM), the clinical isolate of *P. oleovorans* (MICs = 0.4 µM), and the strains belonging to the *P. fluorescens* species (0.8–1.6 µM). Note that *P. fluorescens* and *P. syringae* in iron-deficient environments also excrete the pyoverdine pigment like the yellow and yellowish-green *P. aeruginosa* isolates resistant to G5-PDK, but the *P. fluorescens* in this study were all environmental isolates as *P. syringae*, which are unlikely to experience iron deficiency, and consequently rarely produce pyoverdine (as confirmed by the absence of pigment in the colonies of the strains we tested). Consequently, in the absence of this cationic pigment, the action of G5-PDK on the bacterial membrane was not hindered, and very low MICs were observed. G5-PDK exerted remarkable antibacterial activity against all environmental isolates of *P. straminea* and *P. fragi* (MICs = 1.6 µM), which do not produce pigments. As for *P. putida*, slightly higher MICs were observed for clinical isolates (MICs = 3.2–6.4 µM), while the environmental isolate (strain SMA1) showed a high susceptibility to G5-PDK, displaying a very low MIC (0.8 µM). The different susceptibility to G5-PDK of the strains isolated from the hospitalized individual, compared to that of the strain of environmental origin, can probably be explained by their different lifestyle habits. Indeed, human opportunistic pathogens are used to compete with the host for iron, thus being used to abundantly secrete pyoverdine, as shown in Figure 4 for strain 262. In this way, they can also become less susceptible to the action of G5-PDK, with the consequent increased MIC values found in this study.

To our knowledge, case studies relating to the effects of unconventional antibacterial agents on species other than *P. aeruginosa* and *P. syringae* are very limited. However, we found a study in which the antibacterial and bactericidal properties of isoeugenolo (IE), a 50:50 mixture of ε-polylysine (ε-PL) with dextrin (DX), and mixtures of IE and ε-PL DX were tested, among other bacteria, on *P. fluorescens* (DSMZ 4358) and *P. putida* (DSMZ 291) [80]. While IE displayed very high MICs against both species, the MICs of ε-PL were 6.2 µM against *P. fluorescens* and 7.6 µM against *P. putida*.

Nevertheless, G5-PDK was 1.2–2.4-fold more potent than ε-PL against *P. putida* and 3.9–7.8 times against *P. fluorescens*. Furthermore, regarding the isolates of *P. aeruginosa* producing pyomelanin, we have not found any study to compare the antibacterial activity of G5-PDK against these strains with that of other antibacterial agents. Compared to non-pigment-producing *P. aeruginosa* strains, G5-PDK (MIC = 1.6–6.4 µM) was in most cases more potent than two types of polyionenes, namely 2a and 2b synthesized by Weiyang et al. (MICs = 6.0 µM (2a) and 48.4 µM (2b)) [81]. Regarding the antibacterial activity of G5-PDK against the ATCC *P. aeruginosa* strain (MIC = 6.4 µM), it was 19–156-fold more active than 12 well-characterized cationic polymers (DMAEA, DEAEA, and MEA polymers having either C2 or C12 carbon side chains) (MICs = 122–1000 µM) and several cationic macromolecules molecules belonging to library of 24 compounds [82].

Regarding *P. syringae*, the group of Menkissoglu-Spiroudi prepared a library of 23 hypervalent iodine compounds belonging to aryliodonium salts, aryliodonium ylides, and (diacyloxyiodo)arenes, which were tested for their antibacterial activities against this species. According to the authors, all the compounds examined caused a dose-dependent decrease in bacterial growth rates, and aryliodonium salts, with electron-withdrawing groups, were defined to have higher antibacterial activities (MIC = 8–16 ppm) [83]. According to the observed MICs of G5-PDK towards *P. syringae* (0.2 µM) and those reported by Menkissoglu-Spiroudi et al. expressed in a comparable unit of measurement, their most active compounds (MICs = 17.7–20.1 µM) were 88.5-100.5-fold less active than G5-PDK. Regarding the synthetic cationic macromolecules acting as membrane destroyers, the antibacterial behavior of two self-assembly amphiphilic cationic peptides (R3F3 and R4F4), containing phenylalanine (F) as hydrophobic fraction and arginine (R) as cationic moiety, have recently been investigated and reported [84]. Although both peptides were poorly active against *E. coli* and inactive against *S. aureus*, they were considered active against *P. aeruginosa.* To verify the specificity of the peptides for the *Pseudomonas* genus, the authors also tested R3F3 and R4F4 on other *Pseudomonas* species, including *P. putida, P. agarici, P. fluorescens,* and *P. syringae*. According to the results reported for R4F4, which was the most potent peptide, the MIC value against *P. aeruginosa* PA01 was, in our opinion, extremely high (MIC = 411.1 µM), and too high to consider R4F4 active on this species, although much lower than the IC_50_ that authors have determined for eukaryotic cells (IC_50_ = 2150 µM) [84]. However, considering the pyomelanin-producing *P. aeruginosa* isolates tested in this study, G4-PDK was 64.2–256.9-fold more active than R4F4 against this species, and demonstrated a selectivity index (SI = 12.6–50.5) 2.4–10-fold higher than that calculated for R4F4 (5.2). Survival (%) essays performed on other species showed that R4F4 was particularly active against *P. syringae* (MIC = 8.2 µM) [84]. Accordingly, G5-PDK was 41-fold more potent than R4F4 against this species. Against phytopathogenic bacteria, including *P. syringae*, 7,10-Dihydroxy-8(E)-octadecenoic acid (DOD), a hydroxy fatty acid produced by a particular strain of *P. aeruginosa* species, displayed MIC in the range 125–1000 μg/mL (396–3163 µM), and particularly, against *P. syringae* showed MIC = 500 μg/mL (1580 µM) [85], which denoted 7900-fold lower antibacterial activity than that of G5-PDK. MICs more like those obtained in this study for *P. syringae* were reported by Cameron and co-authors, who treated two strains of *P. syringae* with modified and unmodified dodecapeptide amide molecules (MICs = 3.2–15.4 µM), two hexapeptide amide compounds (MICs = 3.9–7.7 µM), and the 5-nitro-2-furaldehyde (NFA) conjugated hexapeptide acid (MICs = 1.6–3.2 µM). Although the MIC values observed for these compounds showed significant antibacterial activity, G5-PDK was found to be from 8 to 77 times more active [86].

#### 3.2.2. Time-Killing Curves

Time-kill experiments were performed with G5-PDK at concentrations equal to 4 × MIC on strains of *P. fluorescens*, *P. putida* and *P. aeruginosa*. Figure 5 shows the representative killing curves obtained with the environmental isolate *P. fluorescens* (strain SMI6), the clinical isolate pyoverdine-producer *P. putida* (strain 262), and the pyomelanin-producer *P. aeruginosa* (strain 244). G5PDK possessed an extremely strong bactericidal effect against all pathogens tested, as a ≥4 log reduction in the original cell number was observable after 2–4 h, while total eradication of bacterial cells was observed after 6 h of exposure to G5-PDK for *P. fluorescens,* thus establishing a species-dependent variability in the kinetics of time-killing. Significant regrowth was observed for *P. aeruginosa* after 4 h, while after 6 h for *P. putida* and *P. fluorescens* isolates, and at 24 h, a regrowth like that of control strains was observed for all species examined.

These results reproduced those we observed both for the fifth-generation lysine-modified dendrimer G5K [55] and for G5-PDK when it was tested on isolates of the genus *Acinetobacter* [60], where, as in this case, the regrowth was more consistent for *A. baumanni* respect to other *Acinetobacter* species such as *A. pittii* and *A. ursingii*. Furthermore, we have already justified bacterial regrowth by assuming a pH-dependent inactivation of G5-PDK through a process of self-degradation by an intramolecular amidation reaction [60]. This hypothesis might seem in contrast with the results obtained by Hyldgaard et al., who observed a complete reduction of *P. putida* exposed to cationic ε-PL at 2 × MIC (15.2 µM) [80]. Since the structure of ε-PL should undergo the same degradation process as assumed for G5-PDK, a regrowth of *P. putida* had to be equally observable in that case. However, in their study, the authors administered a mixture of ε-PL/dextrin 50/50, in which dextrin, which is a well-known stabilizer, certainly protected ε-PL from self-degradation. In fact, few studies have monitored the bactericidal activity of cationic materials against various bacteria for up to 24 h. In those cases [55,87], while rapid killing occurred after only 5 min [87] or 1 h [55] of exposure of the bacteria to the cationic devices, abundant bacterial regrowth occurred after 24 h of action. However, regrowth of both ATCC strains and clinical isolates of the *P. aeruginosa* genus was also observed when the bacteria were exposed to traditional antibiotics such as ceftazidime, ceftazidime-avibactam, meropenem, and piperacillin-tazobactam [88].

### 3.3. Evaluation of Cytotoxicity of G5-PDK on HaCaT Human Keratinocytes Cells

In general, the ideal drug should have a high toxic concentration, but a very low active concentration. In any research concerning the development of new drugs, an important parameter to determine whether further work on the molecule under consideration can be continued, and whether the new compound could be usable for therapeutic applications, is the selectivity index (SI), which is given by Equation (1) reported in Section 2.3.2. As for new antimicrobial agents, ideally, they should have a high SI, which means low MIC values for bacteria and a high cytotoxic dose for eukaryotic cells.

In this work, to determine the SI value for G5-PDK, we performed dose- and time-dependent cytotoxicity experiments on human keratinocytes (HaCaT).

#### Dose- and Time-Dependent Cytotoxicity Experiments

To evaluate the cytotoxicity of G5-PDK, dose- and time-dependent cytotoxicity experiments were performed on HaCaT keratinocytes cells for the following reasons. *P. aeruginosa,* as the other opportunistic pathogens of *Pseudomonas* considered in the present study, can cause a variety of skin infections, both localized and diffuse [89]. Consequently, to assess the cytotoxicity of G5-PDK, we selected human keratinocytes, which are the primary type of cell found in the epidermis, the outermost layer of the skin, and are more susceptible to colonization by pathogenic bacteria, fungi, parasites, and viruses.

The cytotoxic activity of G5-PDK, as a function of its concentrations (1–100 µM), was determined after 4, 12, and 24 h of exposure of the cells. The results are reported in Figure 6.

As can be seen in Figure 6, after 4, 12, and 24 h of cells exposure, not only did G5-PDK show no cytotoxicity, but at certain concentrations, it was even beneficial for cell growth. In fact, there was a significant proliferation up to concentrations of 50 µM after 4 and 12 h, and of 10 µM after 24 h. Unexpectedly, proliferation at concentrations of 1 and 5 µM was higher than that observed for shorter exposure times. Overall, G5-PDK was only minimally cytotoxic, even at the highest concentration tested (100 µM), which was 16–500 × MIC (83% cell viability) after 4 h of exposure, and was equally insignificantly cytotoxic (83% cell viability) after 12 h of cell treatment at a concentration of 75 µM (12–375 × MIC). After 24 h of exposure, G5-PDK showed no cytotoxicity (95% cell viability) at a concentration of 25 µM (4–125 × MIC), while 69% and 57% of viable cells were observed at concentrations of 50 and 75 µM, respectively (8–250 × MIC and 12–375 × MIC, respectively). Only at a concentration of 100 µM was less than 50%. To get an idea of the selectivity of G5-PDK for bacteria, it was necessary to determine the SI (LD_50_/MIC). Therefore, we determined the LD_50_ from the data obtained from the cytotoxicity experiments. Since cell viability was >80% at all concentrations for 4 h of cell exposure, we could only determine the LD_50_ of G5-PDK for the experiments performed for 12 and 24 h. Firstly, we checked the data of the bar graphs to detect possible outliers, then we converted the bar graphs to dispersion graphs. Using the equations of the mathematical regression models that best suited the dispersions, we calculated the LD_50_ values. Table 5 collects all the concentration and cell viability (%) data obtained from the cytotoxicity experiments performed for 12 and 24 h of exposure.

Figure 7 shows the PCA results obtained working on the data set corresponding to 24 h of exposure. Specifically, Figure 7a shows the PCA results as scores on PC1 vs. PC2, while Figure 7b shows the results as scores on PC2 vs. PC2. Similarly, we also processed the data set corresponding to 12 h of exposure, and the PCA results are available in Appendix A.

As can be observed in Figure 7a,b and as reported in Table 6, data point 6 (5 µM; 131.3%) is very distant from all the other data. Therefore, it was considered as an outlier and was removed before converting the bar graph to a dispersion graph and developing the more correct regression model.

As regards the results obtained by processing the data relating to the 12 h of exposure, the outlier’s values observable in Appendix A and reported in Table 6 were also removed. 

As for the data obtained after 24 h of exposure, based on the value of the coefficient of determination (R^2^), the regression model that best explained the variability of the data was linear, and is shown in Figure 8.

The high value of the coefficient of determination R^2^ (0.9821) ensured the linearity of the regression. The regression equation shown in Figure 4 was used to calculate the LD_50_ and then the SI range, which were included in Table 6. As for the data related to the cytotoxicity experiments performed for 12 h, the regression model that was best suited to the dispersion was the second-degree polynomial (R^2^ = 0.9981, Appendix A), and the related second-degree equation was used to determine the LD_50_ and SI range of G5-PDK for 12 h cell treatments (Table 6). The SI values (24 h) for each strain tested are available in Table 4. Taken together, the SI values of G5-PDK were much higher than those reported as acceptable to consider the new antibacterial agent suitable for therapeutic uses. Furthermore, the SI values at 12 h of exposure were very similar to those obtained for 24 h of exposure, establishing that the effects of G5-PDK marginally depended on the exposure time. As regards the SI values that can be considered satisfactory, the opinions reported are conflicting. In general, some authors have hypothesized that SI values ≥10 make a molecule worthy of further investigation [90,91], while Weerapreeyakul et al. [92] proposed a lower SI value (≥3) to define a clinically applicable molecule as an anticancer agent. In microbiology, Adamu et al. [93,94] reported the antibacterial activities and SI of South African plant leaf extracts, and the most active extract showed an SI of 5.2. Famuyide et al. [95], who described the antibacterial activity of plant extracts on some Gram-positive and Gram-negative bacteria, stated that the extracts could be considered bioactive and non-toxic if SI > 1, while Nogueira and Estolio do Rosario reported that SI should not be less than 2 [96]. Due to these many diverging opinions on the SI acceptance criterion, we believe that further studies are needed to determine the minimum acceptable SI value, but that G5-PDK, possessing SI values (24 h) 1.3–7.8-fold higher than the highest (10) reported as acceptable, it can be considered suitable for clinical development.

## 4. Conclusions

In this study, an autobiodegradable cationic dendrimer containing lysine (G5-PDK), which had previously demonstrated remarkable antibacterial effects against several species of the genus *Acinetobacter*, including MDR *A. baumannii* isolates, was tested on several isolates of *Pseudomonas* genus and on eukaryotic cells (HaCaT) to evaluate its possible future clinical application as an antibacterial agent. In our study, we detected a powerful antibacterial compound (MICs = 0.2–6.4 µM), highly selective for bacteria and non-cytotoxic towards mammalian cells. In addition to exhibiting MICs often lower than those previously reported for cationic antibacterial agents, G5PDK was able to kill both environmental and clinical isolates of *Pseudomonas* species, including *P. aeruginosa*, regardless of their antibiotic resistance. Overall, the novelty of our research consists of having successfully tested G5-PDK not only on nosocomial isolates of *P. aeruginosa* (the main pathogenic species for humans), but also on numerous other species of the genus *Pseudomonas* rarely or never considered in the literature. Furthermore, we have highlighted a possible correlation between the production of some pigments and the greater or lesser tolerance of some isolates of these species to the cationic antibacterial agents such as G5PDK.

Although the *P. fluorescens* and *P. putida* tested in this study were not MDR but minimally resistant, isolates of both species could be responsible for lethal infections, are capable of developing MDR, and could act as reservoirs of resistance determinants that can be spread to other human pathogenic bacteria. Moreover, even if is not pathogenic, *P. fragi* is one of the main culprits behind the spoilage of meat and the deterioration of raw milk, with a notable influence on their shelf life, organoleptic quality, and customer satisfaction. In this regard, a possible use of G5-PDK could effectively help to limit the serious human infections sustained by these species, the spread of multi-resistance, and the waste of food. *P. syringae,* also considered in this work, is responsible for serious plant infections that devastate important crops including kiwi, tomatoes, and tobacco, causing severe economic loss to related industries in Italy, South Korea, Spain, New Zealand, and other countries. Therefore, the development of new antibacterial agents effective against these species is considered urgent for many potential areas of use. Unfortunately, an overview of the available or recently proposed antibacterial treatments highlighted their very low antibacterial activity, which still required a very high concentration to achieve. Interestingly, G5-PDK displayed MICs (0.2 µM) 8–7900-fold lower than those reported for some cationic antimicrobial peptides taken as reference.

Overall, although experiments to investigate if the coadministration of G5-PDK with dextrin as a stabilizer can protect G5-PDK from self-degradation, thus maximally inhibiting bacteria regrowth and improving its bactericidal performances, G5-PDK could already represent a promising new antibacterial agent to limit several species of *Pseudomonas,* which, due to its low cytotoxicity and high SI, could also be used for clinical applications.

## Figures and Tables

**Figure 1 pharmaceutics-13-01411-f001:**
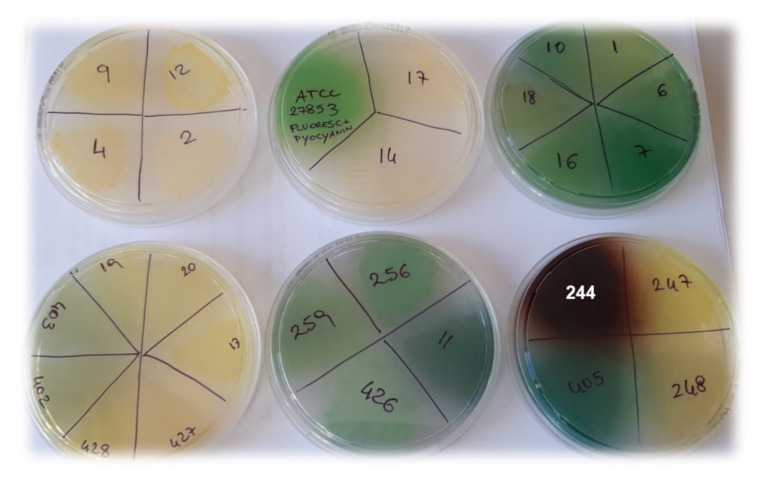
Pigments developed by *P. aeruginosa* isolates, incubated for 48 h at 37 °C on MH agar plates.

**Figure 2 pharmaceutics-13-01411-f002:**
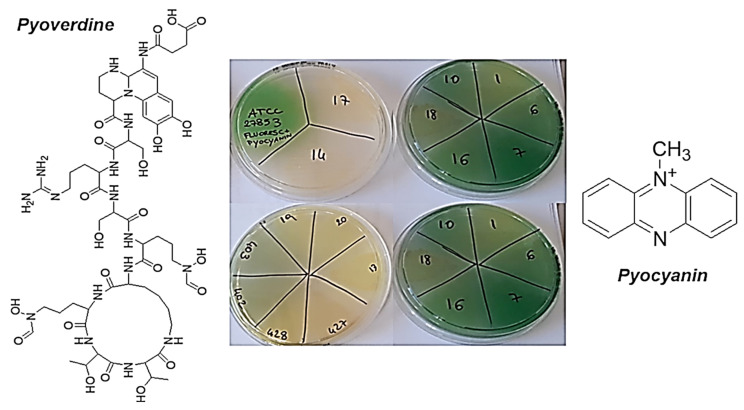
Colors of the colonies that produced pyoverdine, pyocyanin, or a mixture of the two pigments, and their chemical structures.

**Figure 3 pharmaceutics-13-01411-f003:**
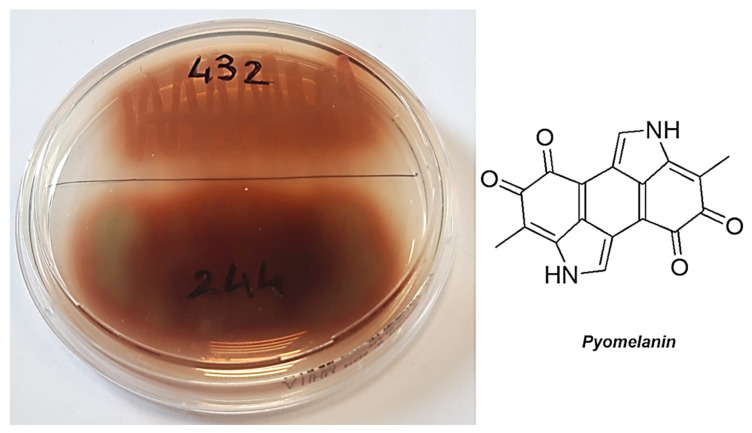
Colonies producing pyomelanin and its chemical structure.

**Figure 4 pharmaceutics-13-01411-f004:**
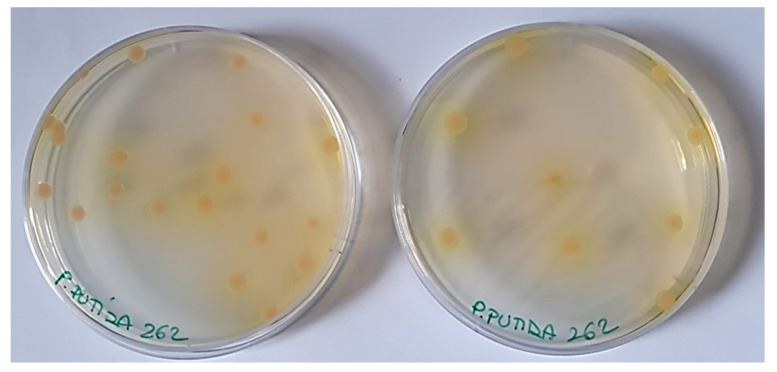
Colors of colonies of *P. putida* 262 clinical isolate producing pyoverdine.

**Figure 5 pharmaceutics-13-01411-f005:**
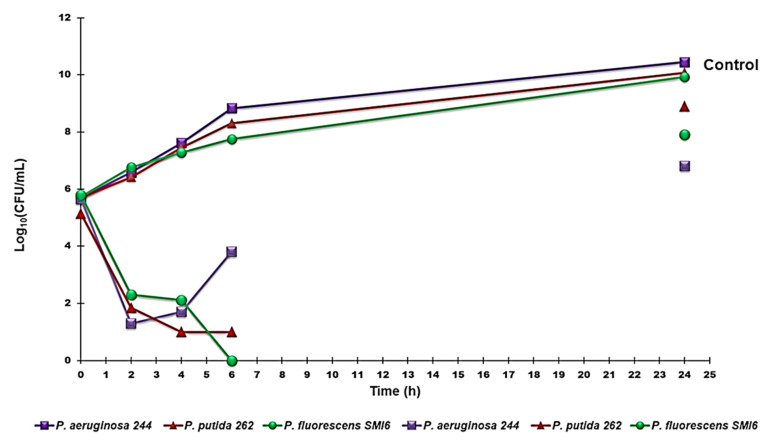
Time-killing curves performed with G5-PDK (at concentrations equal to 4 × MIC) on *P. aeruginosa* 244, *P. fluorescens* SMI6, and *P. putida* 262.

**Figure 6 pharmaceutics-13-01411-f006:**
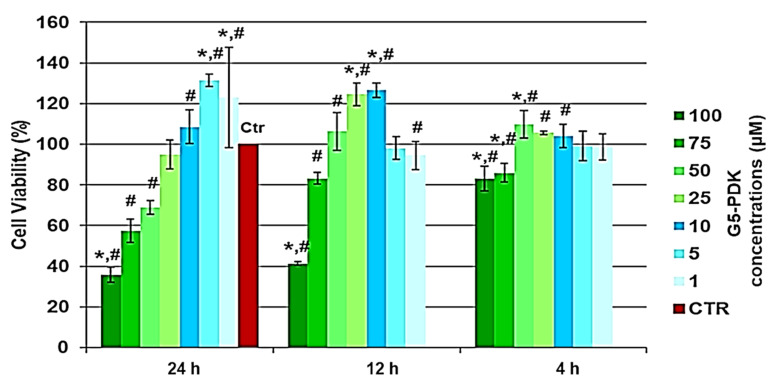
Results of dose- and time-dependent cytotoxicity experiments conducted on HaCaT cells. * *p* < 0.05 with respect to Ctr cells, # *p* < 0.05 between determinations.

**Figure 7 pharmaceutics-13-01411-f007:**
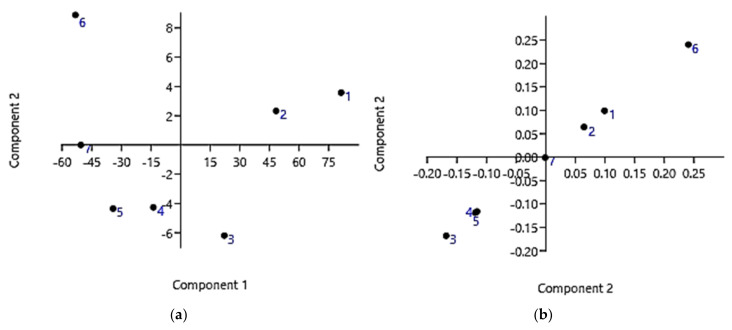
PCA results on data covering 24 h of cells exposure: score plot showing data locations of PC1 vs. PC2 (**a**); score plot showing data locations of PC2 vs. PC2 (**b**).

**Figure 8 pharmaceutics-13-01411-f008:**
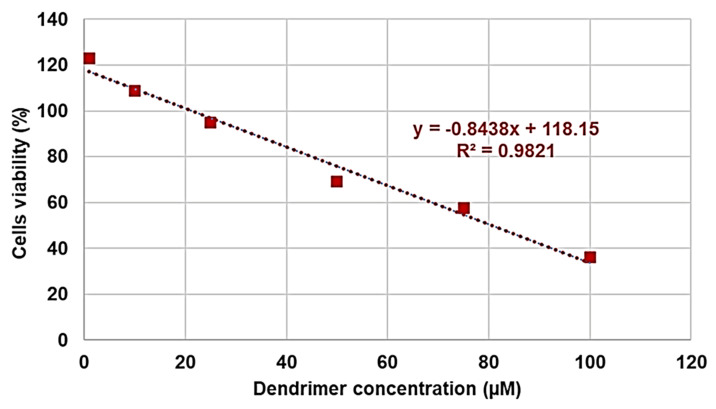
Linear regression model obtained from data relating to 24 h of cells exposure.

**Table 1 pharmaceutics-13-01411-t001:** Main physicochemical features of G5-PD-OH and G5-PD-BK.

Analysis	G5-PD-OH		G5-PD-BK	
FTIR (cm^−1^)	OH C-H alkyl C=O esters	3436 2936 1737	NH C-H alkyl C=ONH urethane -C=O esters NH banding	3380 2870, 2932, 2979 1710 1747 1727
^1^H NMR (400 MHz, DMSO-*d6*) (ppm)	CH_3_ G1-G5, s, 186H CH_2_ PD, m, 2H CH_2_OH, dd, 128H CH_2_O PD, m, 2H CH_2_O PD, m, 2H CH_2_O DR, m, 120H OH, br s, 64H	1.01, 1.16, 1.18, 1.23, 1.34 1.70 3.52 3.56 3.98 4.08–4.18 4.37	CH_3_ G1-G5, CH_2_O PD, CH_2_CH_2_CH_2_lys, m, 572H CH_3_ Boc, s, 576H CH_3_ Boc, s, 576H CH_2_NH lys, m, 128H CH_2_O PD, m, 2H CH_2_O DR + PD + CHNH lys, m, 314H ^α^NHBoc + ^ε^NHBoc lys, m, 128H	0.95–1.90 1.43 1.44 3.10 3.56 4.25 4.70–5.50
^13^C NMR (100 MHz, DMSO-*d6*) (ppm)	C=O G1-G5 CH_2_O G1-G5 Quaternary C G5 Quaternary C G1-G4 CH_3_ G1-G5	173.94, 171.73 64.27, 63.55 50.13 46.12 17.05, 16.61	C=O esters C=ONH urethane Quaternary C Boc CH_2_O G1-G5 CHNH Quaternary C G1-G5 CH_2_NH CH_2_ lys CH_3_ Boc CH_3_ G1-G5	172.32 156.17, 155.63 79.80, 79.02 65.60–65.41 53.37 46.42 40.04 31.84, 29.57, 22.57 28.47, 28.36 17.90–14.20
Elemental Analysis	C, H	51.71, 7.01 * 51.67, 6.98 ^§^	C, H, N	56.78, 8.30, 6.00 * 56.76, 8.18, 6.34 ^§^

G = generation; s = singlet; m = multiplet; d = doublet; dd = double doublet; br s = broad singlet; PD = propandiol; DR = dendrimer; * found; § required; Boc = *tert*-butyloxycarbonyl protecting group.

**Table 2 pharmaceutics-13-01411-t002:** Main physicochemical features of G5-PDK.

Analysis	G5-PDK		
FTIR (cm^−1^)	NH_3_^+^ -C=O esters NH banding	3431 1744 1635	
^1^H NMR (400 MHz, DMSO-*d6*) (ppm)	CH_3_ G1-G5 + CH_2_CH_2_CH_2_ lys, m, 570H CH_2_ PD, m, 2H CH_2_NH_3_^+^ Lys, m, 128 CH_2_O PD, 3.56, 2H, CHNH_3_^+^ lys, m, 64H CH_2_O PD DR + CHNH_3_^+^ lys, m, 250H ^α^NH_3_^+^ lys, br s,192H ^ε^NH_3_^+^ lys, br s, 192H	1.03–1.99 1.70 2.76 3.56 3.99 4.10–4.50 8.20 8.82	
^13^C NMR (DMSO-*d6*, 100 MHz) (ppm)	CH_3_ G1-G5 CH_2_CH_2_CH_2_ lys CH_2_NH_3_^+^ Quaternary C G1-G5 CHNH_3_^+^ CH_2_O G1-G5 C=O	19.33 23.14, 28.01, 31.01 40.02 47.70 53.55 67.65–67.82 170.68–173.33	
Elemental Analysis	C, H, N	41.78, 7.30, 9.10, 22.11 * 41.56, 7.00, 8.90, 22.53 ^§^	
^1^H NMR	MW	20145.3 19961.2 ± 480.2	
Volumetric titration			
DLS ^1^ Analysis	Z-Ave ^2^ (nm) PDI ^3^ Z-potential ^4^ (ζ-p)	203.0 ± 2.6 ^5^ 0.282 ± 0.028 ^5^ +19.2 ± 7.3	
Potentiometric Titration	Max dpH/dV ^5^ HCl 0.1N (mL) ^6^ pH ^7^	10.75 0.6 6.85	4.0 1.2 4.80

G = generation; s = singlet; m = multiplet; d = doublet; dd = double doublet; br s = broad singlet; PD = propandiol; DR = dendrimer; * found; § required; ^1^ dynamic light scattering; ^2^ particle size; ^3^ hydrodynamic diameter of particles; ^4^ polydispersivity index; ^5^ measure of the electrical charge of particles suspended in the liquid of acquisition (water); ^6^ max values of first derivative curve of titration curve, indicating the existence of a two-step protonation process; ^7^ volumes of HCl 0.1N needed to protonate G5-PDK.

**Table 3 pharmaceutics-13-01411-t003:** MIC values of G5-PDK on Gram-negative bacteria of *Pseudomonas* and *Stenotrophomonas* genera obtained from experiments carried out in triplicate ^1^, expressed as µM and as µg/mL.

Strains	MIC (µM (µg/mL))
*P. aeruginosa*	>6.4 (>128)
*P. fluorescens* SMI1	0.8 (16)
*P. putida* SMA1	0.8 (16)
*S. maltophylia*	>25.4 (>512)

^1^ The degree of concordance was 3/3 in all the experiments and the standard deviation (±S.D.) was zero; *P. aeruginosa* and *S. maltophylia* were MDR bacteria.

**Table 4 pharmaceutics-13-01411-t004:** MIC values of G5-PDK obtained on clinical and environmental isolates of *Pseudomonas* genus from experiments conducted in triplicate ^1^ (expressed as µM and as µg/mL), and the selectivity indices (SIs) of G5-PDK for each strain determined using the LD_50_ of G5PDK obtained from the cytotoxicity experiments.

Strains	MIC µM (µg/mL)	Selectivity Index ^2^
*P. aeruginosa* 1 * ^3^	>6.4 (>128)	<13
*P. aeruginosa* 2 # ^3^	6.4 (128)	13
*P. aeruginosa* 4 # ^3^	6.4 (128)	13
*P. aeruginosa 7* * ^3^	>6.4 (>128)	<13
*P. aeruginosa* 9 # ^3^	6.4 (128)	13
*P. aeruginosa* 10 * ^3^	>6.4 (>128)	<13
*P. aeruginosa* 11 * ^3^	>6.4 (>128)	<13
*P. aeruginosa* 12 # ^3^	6.4 (128)	13
*P. aeruginosa* 13 * ^3^	>6.4 (>128)	<13
*P. aeruginosa* 14 * ^3^	>6.4 (>128)	<13
*P. aeruginosa* 16 * ^3^	>6.4 (>128)	<13
*P. aeruginosa* 17 * ^3^	>6.4 (>128)	<13
*P. aeruginosa* 18 * ^3^	>6.4 (>128)	<13
*P. aeruginosa* 19 * ^3^	>6.4 (>128)	<13
*P. aeruginosa* 20 * ^3^	>6.4 (>128)	<13
*P. aeruginosa* 244 § ^3^	1.6 (32)	51
*P. aeruginosa* 247 * ^3^	>6.4 (>128)	<13
*P. aeruginosa* 248 * ^3^	>6.4 (>128)	<13
*P. aeruginosa* 256 # ^3^	6.4 (128)	13
*P. aeruginosa* 259 * ^3^	>6.4 (>128)	<13
*P. aeruginosa* 402 * ^3^	>6.4 (>128)	<13
*P. aeruginosa* 403 * ^3^	>6.4 (>128)	<13
*P. aeruginosa* 405 * ^3^	>6.4 (>128)	<13
*P. aeruginosa* 426 * ^3^	>6.4 (>128)	<13
*P. aeruginosa* 427 * ^3^	>6.4 (>128)	<13
*P. aeruginosa* 428 * ^3^	>6.4 (>128)	<13
*P. aeruginosa* 432 § ^3^	1.6 (32)	51
*P. aeruginosa* 433 * ^3^	>6.4 (>128)	<13
*P. aeruginosa* 434 * ^3^	>6.4 (>128)	<13
*P. aeruginosa* 435 * ^3^	>6.4 (>128)	<13
*P. aeruginosa* 436 * ^3^	>6.4 (>128)	<13
*P. aeruginosa* ATCC 27853 * ^3^	>6.4 (>128)	<13
*P. fluorescens* A8 # ^4^	0.8 (16)	101
*P. fluorescens* SMM8 # ^4^	1.6 (32)	51
*P. fluorescens* SMI1 # ^4^	0.8 (16)	101
*P. fluorescens* SMI2 # ^4^	0.8 (16)	101
*P. fluorescens* SMI6 # ^4^	1.6 (32)	51
*P. fragi* G2 # ^4^	1.6 (32)	51
*P. oleovorans* # ^3^	0.4 (8)	202
*P. putida* 262 * ^3^	3.2 (64)	25
*P. putida* 407 * ^3^	3.2 (64)	25
*P. putida* 409 * ^3^	3.2 (64)	25
*P. putida* 410 * ^3^	6.4 (128)	13
*P. putida* SMA1 # ^4^	0.8 (16)	101
*P. straminea* A5 # ^4^	1.6 (32)	51
*P. straminea* A7 # ^4^	1.6 (32)	51
*P. straminea* A13 # ^4^	1.6 (32)	51
*P. syringae* # ^4^	0.2 (4)	404

^1^ The degree of concordance was 3/3 in all the experiments and standard deviation (±SD) was zero; ^2^ cytotoxicity was evaluated on human keratinocytes after 24 h of cell exposure to G5-PDK; ^3^ clinical isolates; ^4^ environmental isolates; # non-pigment-producing strains on MH agar plates; * pyoverdine and/or pyocyanin-producing strains on MH agar plates; § pyomelanin-producing strains on MH agar plates.

**Table 5 pharmaceutics-13-01411-t005:** Cell concentration and viability data (%).

Label *	µM	Cell Viability (%) 24 h	Cell Viability (%) 12 h
1	100	35.9454	41.34742
2	75	57.4335	83.17176
3	50	68.9837	106.293
4	25	94.8178	124.5387
5	10	108.6001	126.6832
6	5	131.3011	98.12797
7	1	122.8936	94.55272

* Numbers associated with the coupling of data (µM, %) in Figure 7.

**Table 6 pharmaceutics-13-01411-t006:** Outliers and LD_50_ and SI of G5-PDK (12 and 24 h treatments).

Data Set	Outliers		LD_50_ (µM)	SI
	Lables	Values (µM,%)		
12	6, 7	5, 98.12797 1, 94.55272	95.7	15–478
24	6	5, 131.3011	80.8	13–404

## Data Availability

All data concerning this study are contained in the present manuscript or in previous articles whose references have been provided.

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
