# Peer review of "Bactericidal Activity of Non-Cytotoxic Cationic Nanoparticles against Clinically and Environmentally Relevant Pseudomonas spp. Isolates"

_pharmaceutics, 2021, doi:10.3390/pharmaceutics13091411_

Round 1

Reviewer 1 Report

In this manuscript Schito et al have evaluated antibacterial potential of bis hydroxymethyl propionic acid-based lysine functionalized water-soluble generation 5 cationic non-toxic dendrimer G5-PDK against gram negative species of pseudomonas. In my opinion the study is interesting and acceptable for publication in the journal Pharmaceutics.

Author Response

In this manuscript Schito et al have evaluated antibacterial potential of bis hydroxymethyl propionic acid-based lysine functionalized water-soluble generation 5 cationic non-toxic dendrimer G5-PDK against gram negative species of pseudomonas. In my opinion the study is interesting and acceptable for publication in the journal Pharmaceutics.

It is with great pleasure that we thank the Reviewer for his positive comments concerning our study.

Reviewer 2 Report

The manuscript presents a very promising dendrimer, G5-PDK, which shows low MICs in P. aeruginosa  P. fluorescens and P. putida, compared to the products developed in other reported works. In addition, it shows a relatively high SI >10.

However, we still need to note:

  1. In the introduction, the authors used a large space in introducing different types of Pseudomonas. Little was discussed in current situation and update of antibiotics, which seems to be important given this manuscript is about the application of their cationic dendrimers.
  2. In 2.3.1, why were human skin keratinocytes cells used here? Please provide a simple rationale.
  3. In the scheme S1 in Supporting information, it will be clearer if a 2X can be put in front of D5-A-COOH.
  4. Please indicate what first derivative line of the titration curve in fig. S2
  5. In line 362, the aromatic amine group in phenazine is not as basic, or even weak as the amines in amino acids. So, the reason given here will be not very scientifically strong. Plus, the refs do not provide experimental data to support this. Please use a table to compare the MICs between pyoverdine producing bacteria and pyocyanin bacteria.
  6. In line 486-489, have you used to use dextrin as a stabilizer or other stabilizer to test if this can also protect G5-PDK from self-degradation, so that the bacteria regrowth can be maximally inhibited. If not, I think, this could also be added to in the discussion in future plans following line 584.
  7. Though the manuscript states “Both fluorescens and P. putida, responsible for lethal infections, are capable of developing MDR and acting as reservoirs of resistance determinants that can be spread to other human pathogenic bacteria” (line 603-605), the author did not have a positive control in the experiments to show the bacteria they used are drug resistant.

Author Response

The manuscript presents a very promising dendrimer, G5-PDK, which shows low MICs in P. aeruginosa  P. fluorescens and P. putida, compared to the products developed in other reported works. In addition, it shows a relatively high SI >10.

However, we still need to note:

  1. In the introduction, the authors used a large space in introducing different types of Pseudomonas. Little was discussed in current situation and update of antibiotics, which seems to be important given this manuscript is about the application of their cationic dendrimers.

We agree with the Reviewer that large part of the Introduction has been dedicated to presenting the various species of the genus Pseudomonas considered in our study, but it seemed only right to do so as most of them are little known. However, as rightly suggested by the Reviewer, an additional part concerning the current situation and update of antibiotics and the relative reference have been added. Please, see lines 144-163 and the new Ref. 57,58 and 59.

  1. In 2.3.1, why were human skin keratinocytes cells used here? Please provide a simple rationale.

The requested rational has been provided and included in the main text in the discussion section 3.3.1. Please, see lines 559-561.

  1. In the scheme S1 in Supporting information, it will be clearer if a 2X can be put in front of D5-A-COOH.

As requested in Scheme S1 (SI) a “2 x” has been inserted in front of the structure of D5-A-COOH.

  1. Please indicate what first derivative line of the titration curve in fig. S2

We hope to have understand the request of the Reviewer. So, we kindly make note to the Reviewer that an indication of what line of the two presents in Figure S2 is the first derivative is already present in the legend of the graph and the caption of the Figure. Furthermore, the formula of the first derivative is reported in Table S1, first row, last column. Anyway, an explanation of what the first derivative is has been added in the caption of Figure S2.

  1. In line 362, the aromatic amine group in phenazine is not as basic, or even weak as the amines in amino acids. So, the reason given here will be not very scientifically strong. Plus, the refs do not provide experimental data to support this. Please use a table to compare the MICs between pyoverdine producing bacteria and pyocyanin bacteria.

We kindly note to the Reviewer that we have not asserted that the amine group of phenazine is basic, but that pyocyanin is a cationic phenazine. Indeed, as reported in Ref. 76 (revised manuscript), according to the biosynthesis of pyocyanin (by P. aeruginosa), it is produced in its cationic form. Please consider the Scheme of biosynthesis of pyocyanin in Ref. 76.

Anyway, the doubt of the Reviewer is justified, because pyocyanin can have two resonance limit formulas, the cationic one (which is that formed in biosynthesis and present at physiological pH) and the neutral one, which, incorrectly we have reported in Figure 2. We therefore thank the Reviewer for having raised this issue. Now, the structure of pyocyanin in Figure 2 has been corrected. Also, for clarity, additional explanations have been added in the text (lines 406-412). Concerning the request of the Reviewer to use a Table to compare the MICs between pyoverdine producing bacteria and pyocyanin bacteria, since usually bacteria, even if in different concentrations, produce both pigments, to divide them into two categories would be impossible without committing great errors of assessment. Furthermore, concerning P. aeruginosa, as reported in Table 4, the MICs of all bacteria producing pyoverdine and/or pyocyanin were identical and > 6.4 µM.

  1. In line 486-489, have you used to use dextrin as a stabilizer or other stabilizer to test if this can also protect G5-PDK from self-degradation, so that the bacteria regrowth can be maximally inhibited. If not, I think, this could also be added to in the discussion in future plans following line 584.

We agree with the Reviewer and thank him for his rational suggestion. Then, the additional part requested has been added in the Conclusions section (lines 683-688).

  1. Though the manuscript states “Both fluorescens and P. putida, responsible for lethal infections, are capable of developing MDR and acting as reservoirs of resistance determinants that can be spread to other human pathogenic bacteria” (line 603-605), the author did not have a positive control in the experiments to show the bacteria they used are drug resistant.

Indeed, concerning clinical isolates of P. putida, the antibiograms evidenced drug resistance even if MDR, while P. fluorescens strains, which were all environmental isolates, were not resistant. Anyway, with the sentence indicated by the Reviewer, we did not refer to strains of our study but generally to isolates of P. putida and P. fluorescens species. For clarity, additional explanations have been added in lines 663-666.

Reviewer 3 Report

This manuscript describes the bactericidal activity of biodegradable cationic dendrimer G5-PDK against clinically and environmentally relevant Pseudomonas isolates. The topic is of scientific interest due to the necessity of new generation antibacterial materials to treat the bacterial infections. The manuscript is prepared well with detailed methodology and the results section. I suggest the following points for preparing the final draft.

  1. Page 8, lines 501-506: Looks like methodology. Try to move to the materials and methods section.
  2. In page 10, lines 538-542: The software details were included in the methodology. No need to repeat it again here. I think, it is better to describe the methodology and statistical analysis in the materials and method section.

Author Response

This manuscript describes the bactericidal activity of biodegradable cationic dendrimer G5-PDK against clinically and environmentally relevant Pseudomonas isolates. The topic is of scientific interest due to the necessity of new generation antibacterial materials to treat the bacterial infections. The manuscript is prepared well with detailed methodology and the results section. I suggest the following points for preparing the final draft.

  1. Page 8, lines 501-506: Looks like methodology. Try to move to the materials and methods section.

As requested, the part indicated by the Reviewer (slightly modified) has been moved to the Material and Methods section (Section 2.3.2, lines 257-261).

  1. In page 10, lines 538-542: The software details were included in the methodology. No need to repeat it again here. I think, it is better to describe the methodology and statistical analysis in the materials and method section.

As requested, the part indicated by the Reviewer (slightly modified) has been moved to the Material and Methods section (Section 2.3.2, lines 249-257).